# CXXC5 Mediates DHT-Induced Androgenetic Alopecia via PGD_2_

**DOI:** 10.3390/cells12040555

**Published:** 2023-02-09

**Authors:** Yeong Chan Ryu, Jiyeon Park, You-Rin Kim, Sehee Choi, Geon-Uk Kim, Eunhwan Kim, Yumi Hwang, Heejene Kim, Gyoonhee Han, Soung-Hoon Lee, Kang-Yell Choi

**Affiliations:** 1Department of Biotechnology, College of Life Science and Biotechnology, Yonsei University, Seoul 03722, Republic of Korea; 2CK Regeon Inc., B137 Engineering Research Park, 50 Yonsei Ro, Seoul 03722, Republic of Korea

**Keywords:** PGD_2_, CXXC5, Wnt/β-catenin pathway, DHT

## Abstract

The number of people suffering from hair loss is increasing, and hair loss occurs not only in older men but also in women and young people. Prostaglandin D_2_ (PGD_2_) is a well-known alopecia inducer. However, the mechanism by which PGD_2_ induces alopecia is poorly understood. In this study, we characterized CXXC5, a negative regulator of the Wnt/β-catenin pathway, as a mediator for hair loss by PGD_2_. The hair loss by PGD_2_ was restored by *Cxxc5* knock-out or treatment of protein transduction domain–Dishevelled binding motif (PTD-DBM), a peptide activating the Wnt/β-catenin pathway via interference with the Dishevelled (Dvl) binding function of CXXC5. In addition, suppression of neogenic hair growth by PGD_2_ was also overcome by PTD-DBM treatment or *Cxxc5* knock-out as shown by the wound-induced hair neogenesis (WIHN) model. Moreover, we found that CXXC5 also mediates DHT-induced hair loss via PGD_2_. DHT-induced hair loss was alleviated by inhibition of both GSK-3β and CXXC5 functions. Overall, CXXC5 mediates the hair loss by the DHT-PGD_2_ axis through suppression of Wnt/β-catenin signaling.

## 1. Introduction

The hair follicle is one of the few regenerative organs [1], which undergoes a cycle of anagen (growth phase), catagen (regression phase), and telogen (resting phase) [2]. The major features of hair loss include a shortened anagen phase, a rapid transition to the catagen phase, and eventual miniaturization of hair follicles [3]. Dihydrotestosterone (DHT) and prostaglandin D_2_ (PGD_2_) are known as major factors that cause alopecia while inducing hair follicle miniaturization [4]. The PGD_2_ is elevated in alopecia patients and reduces hair regrowth as well as wound-induced hair follicle neogenesis (WIHN) [5,6]. It is known that PGD_2_ inhibits hair growth through its receptor Gpr44 [6]. However, the mechanism of PGD_2_ in hair loss is poorly understood.

The Wnt/β-catenin signaling pathway is one of the most important signaling pathways that regulate hair follicles [7,8]. The Wnt/β-catenin pathway is essential for the development, maintenance, and WIHN of hair follicles [9,10,11]. The decrease in the Wnt/β-catenin signaling is a key feature in the onset of the catagen phase and alopecia [3,12]. Previously, we identified that CXXC-type zinc finger protein 5 (CXXC5) is a BMP signaling target gene and a negative regulator of the Wnt/β-catenin signaling functioning by interacting with Dishevelled (Dvl) [13,14]. Similar to other alopecia factors, Cxxc5 increases in the late anagen phase to induce the catagen phase and the CXXC5 level is elevated in alopecia patients [12]. Since CXXC5 suppresses the Wnt/β-catenin signaling, both hair regrowth and WIHN are increased in *Cxxc5* knock-out mice [12]. Furthermore, treatment with protein transduction domain–Dvl binding motif (PTD-DBM), a peptide that interferes with CXXC5-Dvl binding function [14], promotes hair regrowth and WIHN, as in *Cxxc5* knock-out mice [12]. Although the role of CXXC5 in hair loss has been identified, the upstream event inducing CXXC5 has not yet been elucidated.

In this study, we identified that expressions of prostaglandin D_2_ synthase (Ptgds) and p-smad1/5/9, a transcription factor of BMP signaling, had similar patterns to those of Cxxc5 during the hair cycle. We also found that PGD_2_ induced CXXC5 through BMP signaling in vitro. PTD-DBM treatment or *Cxxc5* knock-down attenuated the suppressive effects of PGD_2_ in vitro. Both PTD-DBM-treated and *Cxxc5* knock-out mice improved hair regrowth ex vivo and in vivo by restoring Wnt/β-catenin signaling and proliferation. Inhibition of CXXC5 function also restored the suppressed WIHN by PGD_2_. Moreover, DHT was revealed to cause alopecia via suppression of the Wnt/β-catenin signaling pathway by activating GSK-3β [15,16], and PTGDS is induced by the androgen receptor (AR) [17,18]. Our study confirmed that DHT increased PTGDS and subsequently induced CXXC5. We also found that DHT-induced alopecia was alleviated by treatment with KY19382, a small molecule mimetic of PTD-DBM which induces hair growth by simultaneous inhibition of the functions of CXXC5 and GSK-3β [19]. Taken together, we identified that DHT and PGD_2,_ the major inducers of androgenetic alopecia [16]_,_ cause hair loss by Wnt/β-catenin signaling suppression via CXXC5 overexpression. Therefore, we suggest that inhibition of both CXXC5 and GSK-3β functions is an effective approach for male pattern androgenetic alopecia (AGA) treatment.

## 2. Materials and Methods

### 2.1. Mice

Wild-type C57BL/6N mice were obtained from Koatech Co. (Gyeonggido, Korea). The generation of the *Cxxc5* knock-out mouse was described previously [20]. *Cxxc5* heterozygous mice were crossed to obtain wild-type and *Cxxc5* knock-out mice. 

### 2.2. Depilation-Induced Hair Cycle Progression

Seven-week-old wild-type mice were anesthetized by intraperitoneal injection of 2,2,2,-tribromoethanol (400 mg/kg, IP; Sigma-Aldrich, St. Louis, MO, USA). The dorsal skins of mice were plucked to induce hair cycle synchronization [21]. After the experiments were completed, the mice were euthanized using CO_2_ gas.

### 2.3. Hematoxylin and Eosin (H&E) Staining

Skin tissues were fixed with 10% formalin overnight at 4 °C. The tissues were paraffinized and sliced into pieces 4 μm in thickness. The slides were deparaffinized and rehydrated. The slides were incubated in hematoxylin for 5 min and eosin for 1 min. The number of follicles was measured by counting the hair follicles in H&E staining images. Dermal thickness was measured by using ImageJ software V1.48.

### 2.4. Immunohistochemistry (IHC)

Skin tissues were fixed with 10% formalin. Paraffin sections were deparaffinized and rehydrated. The slides were microwaved for 15 min in 10 mM sodium citrate buffer for antigen retrieval. The slides were blocked with 5% BSA in PBS. The slides were incubated with the following primary antibodies: rabbit anti-PTGDS (Abcam Cat# ab182141, RRID: AB_2783784, 1:100, Cambridge, UK), rabbit anti-CXXC5 (Abcam Cat# ab106533, RRID: AB_10858571, 1:100), rabbit anti-β-catenin (Abcam Cat# ab16051, RRID: AB_443301, 1:100), rabbit anti-p-smad1/5/9 (Cell Signaling Technology Cat# 13820, RRID: AB_2493181, 1:100, Danvers, MA, USA), mouse anti-CXXC5 (Santa Cruz Biotechnology Cat# sc-376348, RRID: AB_ 10987896, 1:100, Dallas, TX, USA), mouse anti-PCNA (Santa Cruz Technology Cat# sc-56, RRID: AB_628110; 1:500), rabbit anti-fgf9 (Abcam Cat# ab71395, RRID: AB_2103075, 1:200), rabbit anti-cytokeratin 17 (Abcam Cat# ab53707, RRID: AB_869865, 1:400), or mouse anti-AE13 (Abcam Cat# ab16113, RRID: AB_302268, 1:200). After washing with PBS, the slides were incubated with conjugated secondary antibodies: anti-mouse Alexa Fluor 488 (Life Technologies Cat# A11008, RRID: AB_143165, 1:500, Camarillo, CA USA) or anti-rabbit Alexa Fluor 555 (Life Technologies Cat# A21428, RRID: AB_141784, 1:500), and counterstained with 4′,6-diamidino-2-phenylindole (DAPI; Sigma-Aldrich, St Louis, MO, USA). The images were taken using an LSM510 confocal microscope (Carl Zeiss Inc., Oberkochen, Germany). The fluorescence intensity was measured using Zen software V3.1 (Carl Zeiss Inc., Oberkochen, Germany).

### 2.5. Immunoblotting

The tissues or cells were washed with PBS and lysed in RIPA buffer (150 mM NaCl; 10 mM Tris, pH 7.2; 0.1% SDS; 1% Triton X-100; 1% sodium deoxycholate; 5 mM EDTA). The lysates were centrifuged at 15,920× *g* at 4 °C for 30 min. Then, 20 μg of proteins was separated on 10% SDS-PAGE gels and transferred onto PROTRAN nitrocellulose membranes (Schleicher and Schuell Co., Keene, NH, USA). Precision Plus Protein Standards (161-0373, Bio-rad) were used as molecular weight markers. After blocking with 5% skim milk, the membranes, which contained protein bands, were blotted with the primary antibodies: rabbit anti-PTGDS (Abcam, 1:500), rabbit anti-CXXC5 (Abcam, 1:500), mouse anti-β-catenin (BD Transduction Laboratory Cat# 610154, RRID: AB_397555; 1:2000, Lexington, USA), rabbit anti-Erk1/2 (Cell Signaling Technology Cat# 9102, RRID: AB_330744, 1:2000), rabbit anti-p-smad1/5/9 (Cell Signaling Technology, 1:500), mouse anti-α-tubulin (Cell Signaling Technology Cat# 3873, RRID: AB_1904178, 1:2000), mouse anti-PCNA (Santa Cruz Technology, 1:500), rabbit anti-fgf9 (Abcam, 1:1000), rabbit anti-cytokeratin 17 (Abcam, 1:1000), or rabbit anti-p-GSK-3β (S9) (Cell Signaling Technology Cat# 9336, RRID: AB_331405, 1:1000). The membranes were blotted with IgG secondary antibodies: horseradish peroxidase-conjugated anti-mouse (Cell Signaling Technology Cat# 7076, RRID: AB_330924, 1:3000) or anti-rabbit (Bio-Rad Cat# 1706515, RRID: AB_11125142, 1:3000). The blots were visualized using enhanced chemiluminescence (Amersham Bioscience, Buckinghamshire, UK) and a luminescent image analyzer (LAS-4000; Fujifilm, Tokyo, Japan).

### 2.6. Reverse Transcription and Quantitative Real-Time PCR (qRT-PCR)

The RNA was separated by using Trizol reagent (Invitrogen, Carlsbad, CA, USA) following the manufacturer’s instructions. The total RNA (2 μg) was reversely transcribed using reverse transcriptases (Invitrogen) at 42 °C for 1 h. The resulting cDNA was amplified using a reaction mixture containing 10 pmol of the primer set (Bioneer, Daejeon, Republic of Korea) and iQ SYBR Green Supermix (QIAGEN, Hilden, Germany). The primer sequences are described in Appendix A.

### 2.7. Cell Culture

HaCaT cell lines were incubated in Dulbecco’s modified Eagle’s medium (DMEM) (Gibco, Gaithersburg, MD, USA) including 100 U/mL penicillin/streptomycin (Gibco) and 10% (*v*/*v*) fetal bovine serum (Gibco) at 37 °C in a humidified atmosphere of 5% CO_2_. Primary human DP cells were obtained from Dr. Young-Gwan Sung in the Department of Immunology at Kyungpook National University (Daegu, Republic of Korea). Primary human DP cells from passages 2–7 were used in this study. The cells were cultured in Dulbecco’s low-glucose modified Eagle’s medium (Hyclone, Pittsburgh, PA, USA) supplemented with 10% fetal bovine serum (Gibco, Gaithersburg, MD, USA), 1% antibiotic–antimycotic (Gibco, Gaithersburg, MD, USA), 1 ng/mL bFGF (Peprotech, Princeton, NJ, USA), and 5 μg/mL insulin (Gibco, Gaithersburg, MD, USA) at 37 °C in a humidified atmosphere of 5% CO_2_. 

### 2.8. Immunocytochemistry

HaCaT cell lines were seeded in a 12-well plate on coverslips. Cultured cells were washed with PBS and fixed with 10% formalin (Sigma-Aldrich), and then permeabilized with 0.2% Triton X-100. After blocking with 5% BSA in PBS, the cells were blotted with primary antibodies: mouse anti-CXXC5 (Santa Cruz Biotechnology, 1:20) or rabbit anti-β-catenin (Abcam, 1:20) overnight at 4 °C. After washing with PBS, the cells were blotted with appropriate secondary antibodies: Alexa Fluor 488-conjugated goat anti-mouse or Alexa Fluor 555-conjugated goat anti-rabbit (Molecular Probes, Leiden, the Netherlands) for 1 h at room temperature and counterstained with DAPI. Images were taken using an LSM510 confocal microscope (Carl Zeiss Inc.). The fluorescence intensity was quantified using Zen software V3.1 (Carl Zeiss Inc.)

### 2.9. In Vivo Hair Regrowth Test

For evaluations of hair regrowth by PGD_2_ and PTD-DBM, respectively, the back skins of 7-week-old mice were depilated and topically treated with 300 μL of 10 μg PGD_2_ or 10 mM PTD-DBM every other day from 8 days to 12 or 20 days [5]. To measure hair regrowth, regrown hairs were collected by using a hair clipper and weighed by using a precision balance. The hair shaft length was measured by using ImageJ software V1.48. For the hair regrowth test for DHT or KY19382, the back skins of 8-week-old mice were shaved and topically applied daily with 300 μL of 100 μM DHT or 2 mM KY19382 for 42 days.

### 2.10. Wound-Induced Hair Follicle Neogenesis Assay

The dorsal skins of 3-week-old mice received 1 cm^2^ full-thickness wounds and were treated topically with 20 μL of 10 μg PGD_2_ or 10 mM PTD-DBM from 7 days to 17 or 25 days daily [6].

### 2.11. Ex Vivo Vibrissa Follicle Culture

Mouse vibrissa follicles were isolated from 6–8-week-old C57BL/6N mice. After euthanizing the mice with CO_2_ gas, anagen follicles were separated under a stereomicroscope (Nikon, Tokyo, Japan). The separated follicles were cultured in 500 μL DMEM supplemented with 100 U/mL penicillin/streptomycin (Gibco) and 12.5 μg/mL gentamicin (Gibco) in 24-well plates.

### 2.12. Alkaline Phosphatase Staining

For ALP staining of tissues, 20 μm cryosections were dried, then fixed in 4% paraformaldehyde (Wako, Osaka, Japan) for 5 min. After washing with PBS, the slides were incubated with NBT/BCIP solution (NBT/BCIP tablets, Roche Diagnostics, Rotkreuz, Switzerland).

For whole-mount ALP staining, the wounded skin tissues were placed in 5 mM EDTA in PBS. The dermis layer was separated, fixed in 4% paraformaldehyde, washed with PBS, and then incubated in NBT/BCIP solution. The images of the dermis were taken by using a stereomicroscope (Nikon). The number of ALP-positive neogenic follicles was calculated by counting dark blue dots.

### 2.13. Statistical Analysis

In vivo and in vitro experiments were designed to establish randomization, equal size, and blinding. The statistical analyses were performed only for experiments with group sizes (*n*) ≥ 5. All group sizes represent the numbers of experimental independent values used to determine statistical analyses. All statistical data were expressed as means ± standard error of the mean. Comparisons between two unpaired groups were assessed with Student’s *t*-test. For comparisons between multi-group studies, one-way ANOVA with Tukey’s test was performed and post hoc tests were conducted only when the F of ANOVA obtained the required level of statistical significance (*p* < 0.05). There was no significant variance inhomogeneity. Prism software V5.01 (GraphPad, CA) was used for statistical analyses. The statistical significance is shown in the figures as follows: * *p* < 0.05, ** *p* < 0.005, *** *p* < 0.0005, NS: not significant.

## 3. Results

### 3.1. The Expression Patterns of Ptgds Correlate and Inversely Correlate with Those of Cxxc5 and Β-Catenin, Respectively

As revealed by immunohistochemical (IHC) analyses, the expression profile of hair inhibitory factor Cxxc5 is oppositely regulated with that of β-catenin during the mouse hair cycle [12]. The expression patterns of Ptgds are correlated with those of Cxxc5 and inversely correlated with β-catenin, respectively, during the mouse hair cycle (Figure 1a,b and Appendix A). Ptgds and Cxxc5 proteins started to increase in the late anagen phase and reached a maximum level in the catagen phase [5,12]. Contrastively, β-catenin, a hair promotion factor, was decreased in the catagen phase [22]. The correlative expression patterns of Ptgds and Cxxc5 were also confirmed by quantitative RT-PCR (qRT-PCR) and immunoblotting analyses (Figure 1c,d). Overall, Ptgds, which generates the hair growth inhibitory factor PGD_2_, and Cxxc5 correlate in their expression patterns during the mouse hair growth cycle.

### 3.2. PGD_2_ Suppresses Hair Growth via CXXC5

Previous studies found that CXXC5 is induced by BMP signaling in the brain and bone [13,23]. Since the BMP signaling is involved in the catagen phase [24], we examined the relationship between CXXC5 and BMP signaling in hair cells. The p-smad1/5/9, a transcription factor of BMP signaling, showed similar expression patterns to Cxxc5 and Ptgds (Figure 1b and Appendix A). In addition, mRNA expression patterns of *Ptgds* and *Cxxc5* were similar to those of the p-smad target genes, *Hes1* and *Jag1* [25,26] (Figure 1d). Moreover, we confirmed the mRNA and protein expression levels of CXXC5 were increased by treatment of BMP4 in HaCaT cell lines (Appendix A).

To confirm the relationship between PGD_2_ signaling and CXXC5, we tested effects of PGD_2_ treatment on expressions of Cxxc5 and the BMP signaling target genes in HaCaT cell lines. As *HES1* and *JAG1*, the mRNA level of *CXXC5* was increased by treatment of PGD_2_ (Figure 2a). The levels of CXXC5 and p-smad1/5/9 were dose-dependently increased with the decrement in β-catenin and PCNA (Figure 2b–e). To further confirm the induction of CXXC5 by PGD_2_ via BMP signaling, we tested the effects of Noggin, a BMP signaling inhibitor, on the PGD_2_-induced CXXC5 induction. Here, the increased CXXC5 and decreased β-catenin levels by PGD_2_ were mostly abolished, respectively, by inhibiting BMP signaling by Noggin (Appendix A).

To investigate whether PGD_2_ reduces Wnt/β-catenin signaling via CXXC5, we treated with PTD-DBM to inhibit the CXXC5 function. We confirmed that the reduced Wnt/β-catenin signaling and PCNA by PGD_2_ were recovered by PTD-DBM treatment (Figure 2f–i). Correspondingly, the knock-down of *CXXC5* also restored the decreased Wnt/β-catenin signaling with the proliferation (Figure 2j–m). Taken together, these results show that PGD_2_ induced CXXC5 via BMP signaling, and the increased CXXC5 suppressed the Wnt/β-catenin signaling followed by suppression of the hair cell proliferation.

### 3.3. Blockade of CXXC5 Function Recovers Hair Growth Suppressed by PGD_2_

To identify the recovery effects of blockade of CXXC5 function on hair growth suppressed by PGD_2_, we adapted the ex vivo vibrissa follicle culture system [27]. The inhibition in vibrissa elongation by PGD_2_ treatment was restored by treatment of PTD-DBM (Appendix A). The restoration of the PGD_2_-mediated suppressions of β-catenin expression and proliferation by PTD-DBM treatment in vibrissa hair follicles was confirmed (Appendix A). To further confirm the effects of *Cxxc5* knock-out on the decrement in vibrissa length by PGD_2_, we separated the vibrissa follicles from *Cxxc5* wild-type or knock-out mice. Differently from the vibrissa hair follicles obtained from the *Cxxc5* wild-type mice, those from *Cxxc5* knock-out mice were not suppressed in vibrissa follicle elongation by PGD_2_ (Appendix A). The ineffectiveness of PGD_2_ treatment in the growth of vibrissa hair follicles correlated with no effect on β-catenin and PCNA in *Cxxc5* knock-out mice (Appendix A). To further verify the effects of CXXC5 inhibition on hair growth in vivo, we tested the effects of PTD-DBM or *Cxxc5* knock-out on the hair growth suppression by PGD_2._ The PGD_2_-induced hair suppression was recovered with the restoration of the activation of β-catenin and PCNA by treatment of PTD-DBM (Figure 3a–d and Appendix A). We similarly confirmed the recovery effects by the usage of *Cxxc5* knock-out mice (Figure 3e–g and Appendix A). Overall, the effects of PGD_2_ on hair growth suppression are mediated through the induction of CXXC5.

### 3.4. Inhibition of CXXC5 Function Restores WIHN Reduced by PGD_2_

To confirm the restorative effects of inhibiting CXXC5 function in hair follicle regeneration, we used the WIHN experimental model [11]. The growth of neogenic hair follicles, which were suppressed by PGD_2_ [6], was recovered by treatment of PTD-DBM as shown by ALP staining (Figure 4a,b) as well as by hematoxylin and eosin (H&E) staining (Figure 4c). We revealed induction of fgf9 and keratin 17, WIHN markers [28,29] as well as the β-catenin and PCNA by the PTD-DBM treatment in the PGD_2_-mediated suppression of the neogenic hair follicular formation (Figure 4d–f). Corresponding to the effects of PTD-DBM treatment, we confirmed that Cxxc5 knock-out mice restored hair follicle regeneration after wounding (Figure 4g–i). Moreover, β-catenin, PCNA, and WIHN markers were not affected by PGD_2_ in the Cxxc5 knock-out mice (Figure 4j–l). Overall, the inhibition of the CXXC5 function restored the reduction of WIHN by PGD_2_.

### 3.5. Efficient Improvement of the Androgenetic Alopecia by Inhibition of Functions of Both GSK-3β and CXXC5

Previously, DHT was reported to cause male pattern AGA via suppressing Wnt/β-catenin signaling by activating GSK-3β, one of the components of the β-catenin destruction complex [15,16]. Moreover, PTGDS was identified as the target gene for androgen receptors [17,30]. In this study, we identified that CXXC5 was induced by DHT treatment via PGD_2_ (Appendix A). To further characterize DHT-induced changes in the Wnt/β-catenin pathway, we tested the effects of PTD-DBM, valproic acid (VPA), a small molecular inhibitor of GSK-3β which directly activates Wnt/β-catenin signaling [31], and KY19382, a small molecule that inhibits both CXXC5 and GSK-3β functions [19]. In contrast to VPA or PTD-DBM treatment, KY19382 completely restored β-catenin suppression by DHT (Appendix A). Moreover, only KY19382 completely improved vibrissa hair elongation (Figure 5a,b and Appendix A). These results revealed that hair loss caused by DHT was restored by simultaneous inhibitions of CXXC5 and GSK-3β functions.

In addition, we found that KY19382 recovered the DHT-mediated suppression of hair growth by restoration of Wnt/β-catenin signaling and ALP expression (Figure 5c,d and Appendix A). The recovery of the DHT-mediated hair growth suppression by KY19382 was also revealed by hair regrowth assays and further confirmed by ALP and H&E staining (Figure 5e–g). Quantitative measurements of H&E staining indicated that more hair follicles in the KY19382-treated groups progressed to the anagen phase compared to the hair follicles remaining in the telogen, exogen, and kenogen phases in the DHT-treated group (Figure 5h,i). IHC staining showed that KY19382 treatment promoted hair growth as much as that shown by vehicle treatment through recovery of Wnt/β-catenin signaling (Figure 5j and Appendix A). Taken together, DHT-induced hair inhibition could occur via suppression of Wnt/β-catenin signaling by PGD_2_-induced CXXC5.

## 4. Discussion

AGA occurs in both men and women who are genetically susceptible to androgens [32]. One of the main phenomena in AGA patients is an increase in 5α-reductase, an enzyme that converts testosterone to DHT. Elevated DHT causes hair loss in AGA patients [33]. In addition, PGD_2_ and CXXC5, which were found to be increased in scalps of male pattern AGA patients, inhibit hair growth and WIHN [5,6,12]. However, the interrelationship between them has not yet been investigated [34]. In this study, we found that PGD_2_ induced CXXC5 via activation of BMP signaling, and the increased CXXC5 mediated PGD_2_-induced hair loss. We also confirmed that DHT augmented PTGDS and subsequently induced CXXC5. These results suggest a model for an action mechanism for male pattern AGA by DHT involving the suppression of Wnt/β-catenin signaling through the PGD_2_–CXXC5 axis (Figure 6).

In alopecia, the Wnt/β-catenin signaling in hair cells is reduced, but the topical agent minoxidil (MNX) does not affect Wnt/β-catenin signaling [3,35]. Due to the effectiveness of Wnt/β-catenin signaling on hair growth, including neogenic hair growth, Wnt/β-catenin signaling activators including natural products and small molecules have been tested for hair growth [8,31,36,37]. However, the direct Wnt/β-catenin signaling activators have not shown satisfactory results in clinical trials [38,39]. This could be attributed to the functioning of the negative feedback regulator CXXC5 [12]. Previously, we found that CXXC5 is induced by BMP-Smad signaling and interacts with Dvl to suppress Wnt/β-catenin signaling [13,14]. CXXC5 is overexpressed in scalps of male pattern AGA [12], and the male pattern AGA inducers, DHT and PGD_2_, decreased Wnt/β-catenin signaling via the negative regulator, CXXC5. The role of CXXC5 as a mediator for DHT- and PGD_2_-induced male pattern AGA was further indicated by the recovery effects of inhibition of CXXC5’s function, by PTD-DBM or KY19382.

The role of CXXC5 in male pattern AGA is further supported by various studies which have shown that the involvement of Wnt/β-catenin signaling in male pattern AGA is related to DHT; DHT causes male pattern AGA by decreasing Wnt/β-catenin signaling [3]. DHT activates GSK-3β, which down-regulates the Wnt/β-catenin pathway, by inhibiting phosphorylation at Ser-9 of GSK-3β [15]. DHT-induced reduction of Wnt-3a decreases keratinocyte proliferation [16]. Moreover, DHT modulates Wnt agonists and antagonists; decreasing the Wnt agonists, Wnt-10b and Wnt-5a, while increasing DKK-1, a Wnt antagonist [40,41]. In this study, we found that CXXC5 is a mediator for DHT-induced male pattern AGA through PGD_2_. Overall, the Wnt/β-catenin signaling inhibitor CXXC5 plays a role in DHT-induced PGD_2_ signaling activation and subsequent male pattern AGA. Here, we also found that enhanced activation of Wnt/β-catenin signaling by simultaneous inhibitions of GSK-3β activity and CXXC5-Dvl PPI by KY19382 is much more effective than that acquired by single inhibition by VPA or PTD-DBM. Therefore, a chemical approach effectively activating Wnt/β-catenin signaling by blockade of the functions of CXXC5 and GSK-3β provides an ideal approach for the treatment of male pattern AGA.

Although use of the mouse system to investigate the mechanism related to the hair loss and treatments of alopecia involving DHT is not ideal due to the differences between mouse and human systems, studies using mice have been used frequently [42,43,44,45,46]. In addition, mice have also been used in the applicaiton of clinical treatment [38,39]. Overall, considering the role of CXXC5 as an important mediator of DHT action and phenotypical outcomes, the inhibition of CXXC5 function is a potential treatment for androgenetic alopecia.

## Figures and Tables

**Figure 1 cells-12-00555-f001:**
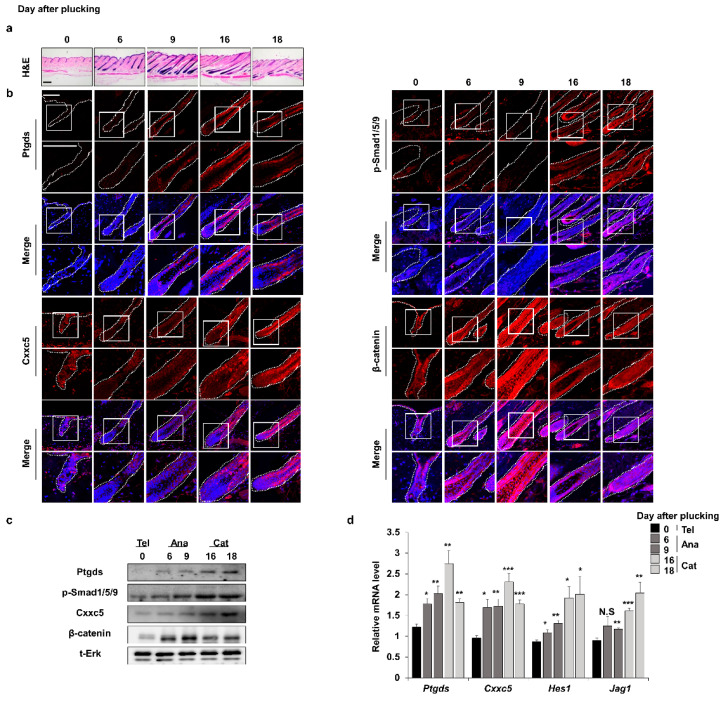
Expression profiles of factors related to hair growth during the depilation-induced hair cycle. The dorsal skins of C57BL/6N 7-week-old wild-type male mice were plucked and harvested at 0, 6, 9, 16, or 18 days. (**a**) H&E staining for dorsal skins. (**b**) IHC staining for Ptgds, p-smad1/5/9, Cxxc5, and β-catenin. (**c**) Immunoblotting assay of whole skin for Ptgds, p-smad1/5/9, Cxxc5, β-catenin, and total Erk. (**d**) qRT-PCR of whole skin for *Ptgds*, *Cxxc5*, *Hes1*, and *Jag1* (*n* = 5). Scale bars = 100 µm. Dashed lines indicate hair follicles. Values are expressed as the means ± SEM. Student’s *t*-test: * *p* < 0.05, ** *p* < 0.005, *** *p* < 0.0005, N.S: not significant.

**Figure 2 cells-12-00555-f002:**
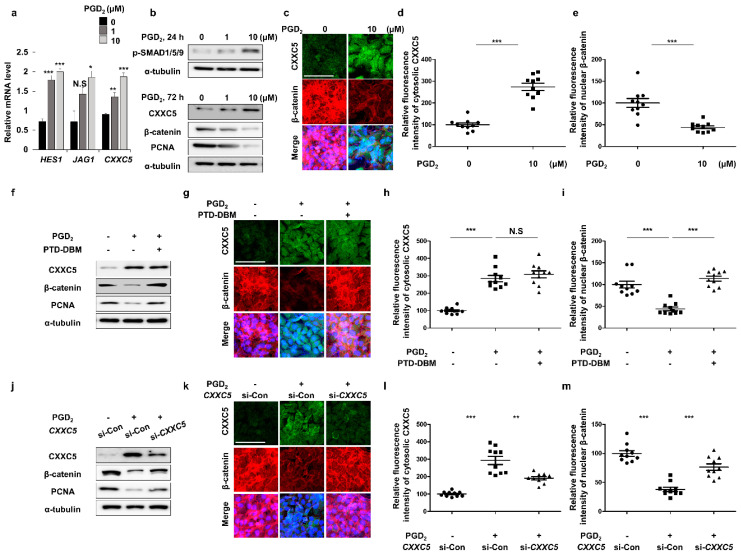
Effects of PGD_2_ and PTD-DBM on hair markers. (**a**–**e**) HaCaT cell lines were incubated with 1 or 10 μM PGD_2_ for 24 or 72 h. (**a**) qRT-PCR analyses for measurement of *HES1*, *JAG1*, or *CXXC5* at 24 h after PGD_2_ treatment. (**b**) Immunoblotting analyses for p-SMAD1/5/9, CXXC5, β-catenin, PCNA, and α-tubulin. (**c**) ICC staining for CXXC5 and β-catenin. (**d**–**e**) Quantitative calculations of fluorescence signal for cytosolic CXXC5 and nuclear β-catenin (*n* = 10). (**f**–**i**) HaCaT cells were incubated with 10 μM PGD_2_ or 10 μM PTD-DBM for 72 h. (**f**) Immunoblotting analyses for PCNA, CXXC5, β-catenin, and α-tubulin. (**g**) ICC analyses for β-catenin and CXXC5. (**h**,**i**) Quantitative measurements of nuclear β-catenin and cytosolic CXXC5 (*n* = 10). (**j**–**m**) HaCaT cells were incubated with 10 μM PGD_2_ or *CXXC5* siRNA transfection for 72 h. (**j**) Immunoblotting analyses of CXXC5, β-catenin, α-tubulin, and PCNA. (**k**–**m**) ICC staining and quantitative calculations for CXXC5 and β-catenin (*n* = 10). Scale bars = 100 µm. Values are expressed as means ± SEM. Student’s *t*-tests: ** *p* < 0.005, *** *p* < 0.0005, N.S: not significant.

**Figure 3 cells-12-00555-f003:**
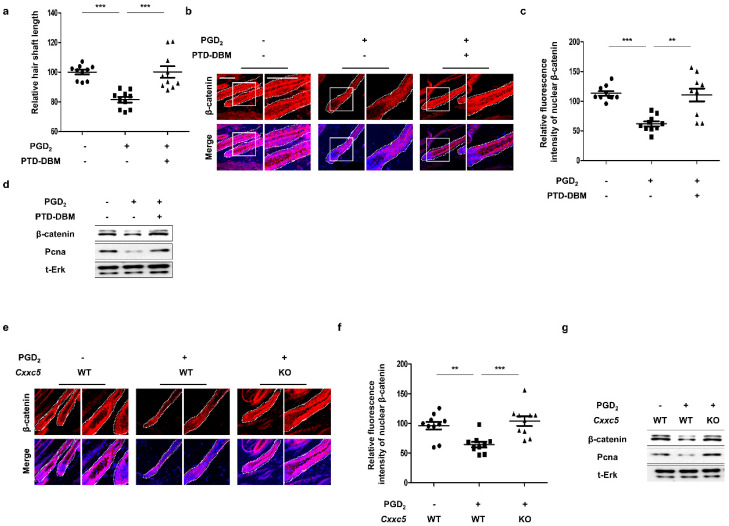
Effects of blockade of Cxxc5 function on hair regrowth. (**a**–**d**) The dorsal skins of C57BL/6N 7-week-old wild-type mice were depilated and harvested at 12 or 20 days. Topical treatment with 10 μg PGD_2_ or 10 mM PTD-DBM every other day was performed from 8 days to 12 or 20 days. (**a**) The relative hair shaft length 20 days after depilation (*n* = 10). (**b**) IHC staining for β-catenin at 12 days after depilation. (**c**) Quantitative calculations for fluorescence intensities of nuclear β-catenin (*n* = 10). (**d**) Immunoblotting analyses for β-catenin, Pcna, and total Erk at 12 days. (**e**–**g**) The back skins of 7-week-old *Cxxc5* wild-type or knock-out mice were depilated and obtained at 12 or 20 days. Treatment of 10 μg PGD_2_ was conducted from 8 days to 12 or 20 days every other day. (**e**) IHC analyses for β-catenin at 12 days. (**f**) Quantitative values for intensities of nuclear β-catenin (*n* = 10). (**g**) Immunoblotting assay for β-catenin, total Erk, and Pcna at 12 days. Scale bars = 100 µm. Dotted lines indicate hair follicles. Values are expressed as means ± SEM. Student’s *t*-tests: ** *p* < 0.005, *** *p* < 0.0005.

**Figure 4 cells-12-00555-f004:**
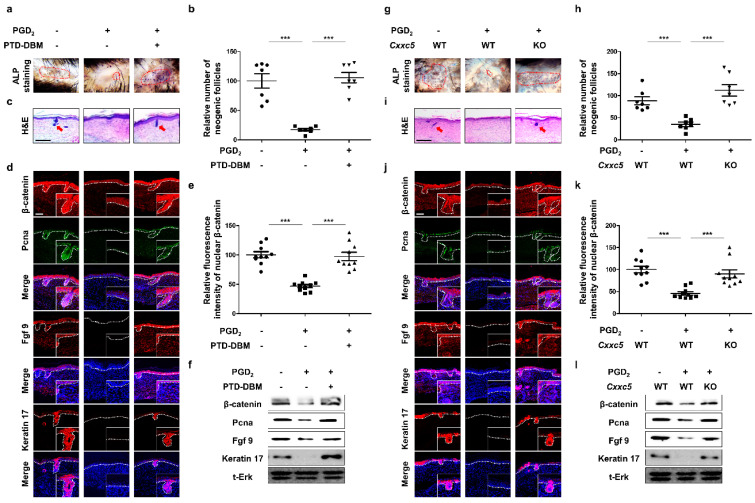
Effects of inhibition of Cxxc5 function on wound-induced hair neogenesis. (**a**–**f**) The dorsal skins of 3-week-old C57BL/6N wild-type mice were wounded and harvested at 21 or 25 days. Topical treatment of 10 μg PGD_2_ or 10 mM PTD-DBM was conducted from 7 days to 21 or 25 days daily after wounding. (**a**,**b**) ALP staining of neogenic follicles 25 days after wounding (*n* = 7). (**c**) H&E staining of wounded skin 21 days after wounding. (**d**) IHC staining for Pcna, β-catenin, keratin 17, and Fgf9 at 21 days. (**e**) Quantitative calculations for nuclear β-catenin (*n* = 10). (**f**) Immunoblotting analyses for β-catenin, Pcna, Fgf9, keratin 17, and total Erk at 21 days. (**g**–**l**) The back skins of *Cxxc5* wild-type or knock-out mice were cut and harvested at 21 or 25 days after wounding. Treatment with 10 μg PGD_2_ was performed from 7 days to 21 or 25 days. (**g**,**h**) ALP staining for neogenic follicles at 25 days (*n* = 7). (**i**) H&E staining of dorsal skins at 21 days. (**j**) IHC analyses for keratin 17, Fgf9, β-catenin, and Pcna at 21 days. (**k**) Quantitative values for nuclear β-catenin (*n* = 10). (**l**) Immunoblotting assay for Fgf9, keratin 17, Pcna, β-catenin, and total Erk at 21 days. Scale bars = 100 µm. Dotted lines represent ALP-positive newly formed hair (**a**,**g**) or the boundary between the dermis and epidermis (**d**,**j**). Arrows indicate neogenic hair. Values are expressed as means ± SEM. Student’s *t*-tests: *** *p* < 0.0005.

**Figure 5 cells-12-00555-f005:**
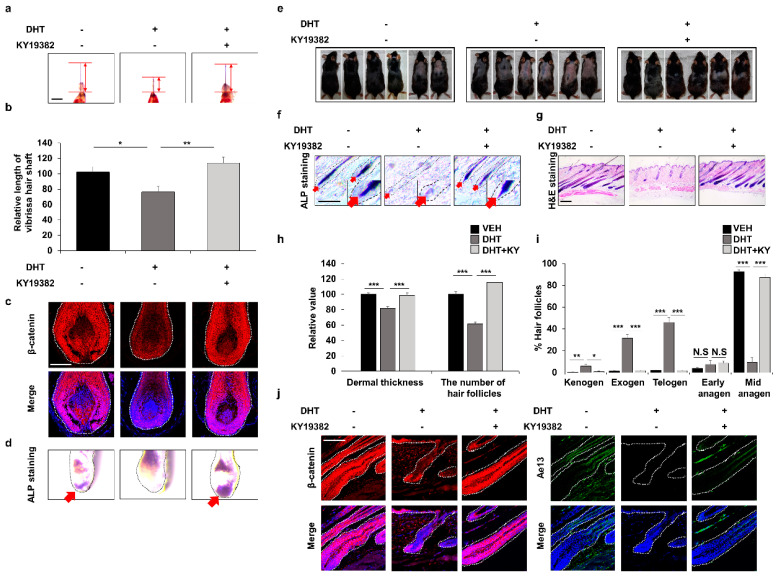
Effects of inhibition of both Cxxc5 and Gsk-3β functions on hair growth of the DHT-treated mice. (**a**–**d**) Vibrissa follicles were separated and cultured with 100 nM DHT or 0.5 μM KY19382 for 4 or 7 days. (**a**,**b**) The relative length of vibrissa follicles at 7 days (*n* = 34). (**c**) IHC staining for β-catenin at 4 days. (**d**) ALP staining for vibrissa follicles. Arrows indicate ALP-positive region. (**e**–**j**) The dorsal skins of C57BL/6N 8-week-old wild-type mice were shaved and topically treated daily with 100 μM DHT or 2 mM KY19382. The dorsal skins were harvested at 42 days. (**e**) Gross images of regrown hair in mice (*n* = 6). (**f**) ALP staining for dorsal skins. (**g**) H&E staining for mouse dorsal tissues. (**h**,**i**) Quantitative measurements of dermal thickness, the number of hair follicles, and hair cycle score using H&E images (*n* = 30). (**j**) IHC analyses for β-catenin and Ae13 using dorsal skins. Scale bars = 100 µm. Dashed lines represent hair follicles. Arrows mean ALP-positive region. Values are expressed as the means ± SEM. Student’s *t*-test: * *p* < 0.05, ** *p* < 0.005, *** *p* < 0.0005, N.S: not significant.

**Figure 6 cells-12-00555-f006:**
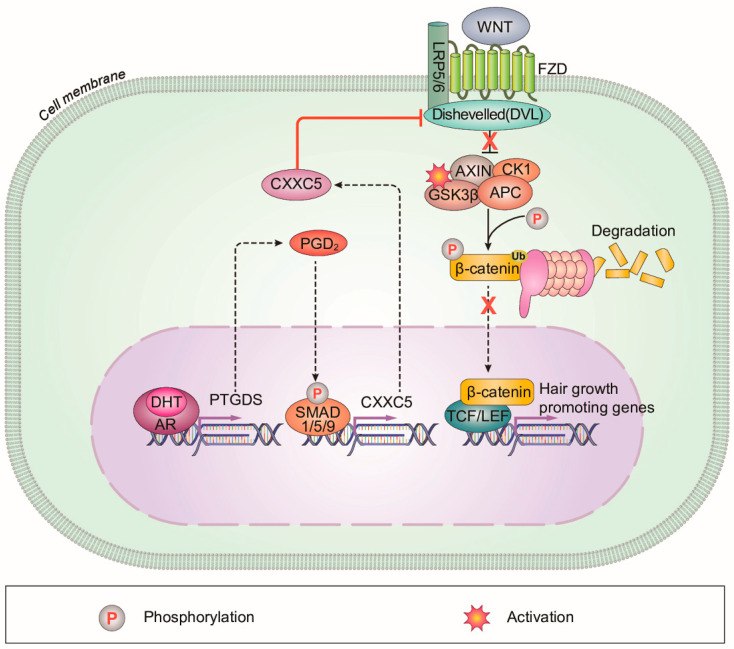
A schematic model for the DHT-induced alopecia mediated via PGD_2_–CXXC5 axis. CXXC5 plays a role as a mediator of DHT-induced androgenetic alopecia via PGD_2_. CXXC5 is induced by DHT-PGD_2_ to suppress the Wnt/β-catenin signaling.

## Data Availability

The data that support the findings of this study are available on request from the corresponding author. The data are not publicly available due to privacy or ethical restrictions.

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
