# Peer review of "CXXC5 Mediates DHT-Induced Androgenetic Alopecia via PGD2"

_cells, 2023, doi:10.3390/cells12040555_

Round 1

Reviewer 1 Report

This manuscript describes the role of CXXC5 in regulation of hair growth. In general, the methodology and results are well designed.

Spell out PTD-DBM in the introduction  section

Avoid repetition of concepts mentioned in the introduction section and in the discussion section

English may be improved.

Author Response

Reviewer 1) Reviewer 1 noted that this manuscript described the role of CXXC5 in regulation of hair growth. In general, the methodology and results are well designed.

The authors appreciate the kind comments from Reviewer 1. To further improve the quality of the manuscript, we carefully revised the manuscript by addressing the following issues suggested by Reviewer 1.

  • Spell out PTD-DBM in the introduction section.

As the reviewer recommended, we described what PTD-DBM stands for in the revised manuscript (Page 3, line 7; page 4, line 19).

  • Avoid repetition of concepts mentioned in the introduction section and in the discussion section.

In order to avoid repetition of explanation in the introduction and discussion section, the repeated descriptions of the discussion section were deleted in the revised manuscript (Page 16, line 13; page 17, lines 1 – 2).

  • English may be improved.

According to the reviewer’s request, we improved many English expressions in the manuscript and reviewer 3 kindly pointed out many wording errors so that the English of the manuscript could be improved.

Reviewer 2 Report

Ryu et al. reported the role of CXXC5 in DHT-induced AGA in vitro and in vivo. The work is of interest. However, some concerns need to be discussed:

1.     All the immunofluorescence staining images failed to display the expression pattern of the proteins. However, the immunofluorescence images in supplementary files are clear. The authors should substitute or reorganize these images to support their claims. For example, the authors claim the nuclear expression of β-catenin, but we cannot see the boundary of cells in Figure 1b.

2.     Original images for blots/gels. The author just provided the final picture in Figures. The original images should be images without any cuts or modifications.

3.     For all the in vivo experiments, it is better to provide gross photos of the mice. Especially the phenotype of hair follicles in CXXX5 KO mice. However, the author only showed gross photos of DHT- or KY19382-treated mice.

4.     The major points in Figure 6 should be described in legends. In addition, no supporting data was found for the transcriptional regulation of CXXC5 by SMAD1/5/9. It should be discussed in the Discussion section if it is from references. It is the same as the relationship between CXXC5 and DVL. It is reported that CXXC5-Dvl protein-protein interaction is the target of KY19382.

Author Response

Reviewer 2) Reviewer 2 stated that the authors reported the role of CXXC5 in DHT-induced AGA in vitro and in vivo. The reviewer mentioned that this study was interesting, but had some concerns to discuss.

The authors thank to the kind comments from Reviewer 2. We addressed the issues below raised by the reviewer.

  • All the immunofluorescence staining images failed to display the expression pattern of the proteins. However, the immunofluorescence images in supplementary files are clear. The authors should substitute or reorganize these images to support their claims. For example, the authors claim the nuclear expression of β-catenin, but we cannot see the boundary of cells in Figure 1b.

As the reviewer advised, we replaced all images that did not adequately show the protein expression patterns, including Figure 1b, with appropriate images.

  • Original images for blots/gels. The author just provided the final picture in Figures. The original images should be images without any cuts or modifications.

Previously, we received an e-mail requesting original images from Cassie guo, section managing editor, and sent the original images, but it seems that they were not delivered. We uploaded the original images again along with the revised file.

  • For all the in vivo experiments, it is better to provide gross photos of the mice. Especially the phenotype of hair follicles in CXXX5 KO mice. However, the author only showed gross photos of DHT- or KY19382-treated mice.

We provided the gross images only for animal experiments showing significant differences in phenotype throughout the paper. Specifically, referring to the description of section 2.9 In vivo hair regrowth test, in the DHT-treated mouse test (Figure 5), the drug was applied for 42 days after shaving the back skins of 8-week-old mice whose hair follicles were in the telogen phase [1]. The hair follicles in the control group progressed to the anagen phase, but the anagen transition was blocked by DHT treatment, and KY19382 reversed the DHT-inhibited anagen transition. The gross images in this experiment were provided due to the significant differences in dorsal skin color showing the hair cycle between the experimental groups. However, in the CXXC5 KO mice experiments (Fig. 3), the gross images were not provided because there was no significant difference in the color of the mouse back skin between the experimental groups due to depilation (Figure R1 in this revision letter).

  • The major points in Figure 6 should be described in legends. In addition, no supporting data was found for the transcriptional regulation of CXXC5 by SMAD1/5/9. It should be discussed in the Discussion section if it is from references. It is the same as the relationship between CXXC5 and DVL. It is reported that CXXC5-Dvl protein-protein interaction is the target of KY19382.

According to the reviewer’s request, we described the major points of Figure 6 in the figure legends (Page 29, lines 9 – 10).

              We also added the description of previous studies that identified CXXC5 as a BMP target gene and suppressor of Wnt/β-catenin signaling by interacting with Dvl in the revised manuscript (Page 17, lines 9 – 10).   

Reviewer 3 Report

My opinion is that this is a thorough and well-written study, I congratulate the authors.

I would like to recommend some wording errors for correction:

- at the end of chapter 1.: „Therefore, we suggest that inhibition of both CXXC5 and GSK-3β functions is an effective approach for AGA treatment.”- it is not stated what term AGA stands for (I understand that it is for androgenetic alopecia but it must be marked once in the text)

- chapter 3.4.: „ALP staning” → „ALP staining”

- chapter 3.5.: „To further characterized role of the Wnt/β-catenin…”- this sounds strange, something should be rephrased

- chapter 3.5.: „that inhibits the both CXXC5 and GSK-3β functions” that inhibits both CXXC5 and GSK-3β functions” 

- chapter 3.5.: „restored by dual inhibition of CXXC5 and GSK-3β…”- this sounds strange, something should be rephrased (maybe „simultaneous inhibition” instead of „dual inhibition”?)

- chapter 3.5.: „hair inhibition could occurs” → „hair inhibition could occur”

- chapter 4.: „In addition, PGD2 and CXXC5 were found to be increased in scalps of the AGA patients, and those inhibit hair growth and WIHN” - this sounds strange, something should be rephrased (maybe: „In addition, PGD2 and CXXC5, which were found to be increased in scalps of the AGA patients, inhibit hair growth and WIHN”)

- chapter 4.: „the topical agent MNX”- it is not stated what term MNX stand for (Minoxidil?)

- chapter 4.: „have been frequently have been used”- it should be rephrased

Author Response

Reviewer 3) Reviewer 3 mentioned that this is a thorough and well-written study. He or she pointed out some wording errors for correction.

We thank the reviewer for the kind comments. As pointed out by the reviewer, we appropriately corrected the following wording errors.

  • “AGA”
  • “androgenetic alopecia (AGA)” (Page 5, lines 15 – 16).

  • “ALP staning”
  • “ALP staining” (Page 14, line 19).

  • “To further characterized role of the Wnt/β-catenin…”
  • “To further characterize DHT-induced changes in the Wnt/β-catenin signaling…” (Page 15, line 12).

  • that inhibits the both CXXC5 and GSK-3β functions” 
  • “that inhibits both CXXC5 and GSK-3β functions” (Page 15, line 15).

  • dual inhibition”
  • “simultaneous inhibitions” (Page 15, lines 19; page 18, line 2).

  • “hair inhibition could occurs”
  • “hair inhibition could occur” (Page 16 line 7).

  • “In addition, PGD2 and CXXC5 were found to be increased in scalps of the AGA patients, and those inhibit hair growth and WIHN
  • “In addition, PGD2 and CXXC5, which were found to be increased in scalps of the male pattern AGA patients, inhibit hair growth and WIHN” (Page 16, lines 14 – 15).

  • MNX”
  • “minoxidil (MNX)” (page 17 lines 3 – 4).

  • “have been frequently have been used”
  • have been used frequently (Page 18, line 9).

Reviewer 4 Report

I think this is an excellent paper.

I would suggest a more detailed hair cycle description adding exogen and kenogen phases.

I think the role of androgen in female pattern alopecia is unclear and the role of DHT appears less significant. There is only a small population of women that respond to a DHT blocker such as finasteride.

Suggest expanding on possible explanations why this latest finding might be more applicable to therapy in humans than previous work on WNT and SHH in mice.

Author Response

Reviewer 4) Reviewer 4 noted that this is an excellent paper.

We thank for the kind comments from reviewer 4.

  • The reviewer would suggest a more detailed hair cycle description adding exogen and kenogen phases..

As suggested by the reviewer, we added quantitative data and description of hair cycle including exogen and kenogen phases (Figure 5i and page 16, lines 2 - 5 in the revised manuscript).

  • The reviewer thought the role of androgen in female pattern alopecia was unclear and the role of DHT appeared less significant. The reviewer noted that there is only a small population of women that respond to a DHT blocker such as finasteride.

We agree with the reviewer’s comments. Since only male mice were used in our experiment, the role of CXXC5 in DHT-induced androgenetic alopecia, but not in female pattern alopecia, was addressed in our study. To make this clear, we added the word ‘male pattern’ before androgenetic alopecia [2] in the revised manuscript (Page 5, line 15; page 15, line 8; page 16, lines 15 and 19; page 17, lines 11, 13, 15, 16, 17 and 22; page 18, lines 1 and 5 – 6).

  • He or she suggested expanding on possible explanations why this latest finding might be more applicable to therapy in humans than previous work on WNT and SHH in mice.

The Wnt/β-catenin signaling pathway plays an essential role in the formation and growth stimulation of hair, and neogenesis of hair follicle [3-6]. The Wnt/β-catenin signaling activators promote hair growth but fail to maintain hair growth and often have marginal effects in clinical trials [7,8]. This limited effect might be caused by the increased expression of Wnt signaling inhibitors such as DKK1 or CXXC5 through the negative feedback regulation following Wnt signaling activation [9,10]. The elevated level of CXXC5 in bald scalps and the efficiency of PTD-DBM or KY19382 on hair regeneration in mice propose the potential usage of KY19382 as a clinical treatment for hair loss. We have described this in the revised manuscript (page 17, lines 3 – 14).

Round 2

Reviewer 2 Report

1. Cxxc5 knock-out mice were not suppressed in vibrissa follicle elongation by PGD2. There was no significant difference in the color of the mouse back skin and hair follicle regeneration between Cxxc5 KO and WT mice. The results are different between in vivo and in vitro data. The authors should discuss it.

2. The immunofluorescence staining images are still not clear. Maybe it is resulted from the conversion of pdf. Please pay attention to it in the next steps.

Author Response

February 3, 2023

Dr. Cassie Guo

Section Managing Editor, Cells

RE: cells-2175774

“CXXC5 mediates DHT-induced androgenetic alopecia via PGD2

Dear Dr. Alison Li

Thank you very much for your letter on February 2, 2023, recommending a revision to our manuscript. We are grateful to the editors and reviewers for their kind comments that our manuscript has been greatly improved through revision. To reflect the suggestions by the reviewers, we carefully addressed all issues raised by the reviewers in a point-by-point manner. We hope you find our revised manuscript is satisfactory.

Thank you again for your kind consideration of our manuscript.

Sincerely yours,

Kang-Yell Choi, Ph.D.

Professor, College of Life Science and Biotechnology, Yonsei University

134 Shinchon-Dong, Seodemun-Gu, Seoul 03722, Korea

E-mail: kychoi@yonsei.ac.kr; Tel: +82-2-2123-7438; Fax: +82-2-2123-8284

* RESPONSES TO REVIEWERS

Reviewer 2)

  • The reviewer mentioned that Cxxc5 knock-out mice were not suppressed in vibrissa follicle elongation by PGD2, whereas there was no significant difference in the color of the mouse back skin and hair follicle regeneration between Cxxc5 KO and WT mice, representing that the results are different between in vivo and in vitro He or she requests to the authors should discuss it.

Thanks for the kind comments from this reviewer. As noted by the reviewer, Cxxc5 deletion reversed the vibrissa hair follicle elongation suppressed by PGD2 (Supplementary Figure S4e and S4f). Consistently, Cxxc5 knockout reversed the hair re-growth inhibited by PGD2 in mice, as evidenced by the weight measurement of re-grown hairs (Supplementary Figure S5c).

We finished the experiment at day 20 after plucking to accurately measure the amount of fully re-grown hair, but at this time, the hair follicles of control mice had already undergone the catagen-to-telogen transition [1], so the hair follicles of PGD2-treated mice similarly revealed at a stage of telogen phase (Figure R1 in this revision letter).

  • The reviewer noted that the immunofluorescence staining images were still not clear. Maybe it resulted from the conversion of pdf. The reviewer suggested that the author pay attention to it in the next steps.

Thanks for the detailed advice. We provide the figure as TIFF files with a resolution of 300 dpi. Nevertheless, if this problem persists, we will send the original files via an e-mail to the editor.

  1. Müller-Röver, S.; Handjiski, B.; van der Veen, C.; Eichmüller, S.; Foitzik, K.; McKay, I.A.; Stenn, K.S.; Paus, R. A comprehensive guide for the accurate classification of murine hair follicles in distinct hair cycle stages. The Journal of investigative dermatology 2001, 117, 3-15, doi:10.1046/j.0022-202x.2001.01377.x.

Figure R1. Effects of Cxxc5 knock-out on the inhibition of hair regrowth by PGD2. The dorsal skins of 7-week-old Cxxc5 wild-type or knock-out mice were depilated and harvested at 20 days. Treatment of 10 μg PGD2 was conducted from 8 days to 20 days every other day. The gross image was photographed after clipping and weighing the re-grown hairs.
